# Biomimicry as a Sustainable Design Methodology—Introducing the ‘Biomimicry for Sustainability’ Framework

**DOI:** 10.3390/biomimetics7020037

**Published:** 2022-03-30

**Authors:** Lazaara Ilieva, Isabella Ursano, Lamiita Traista, Birgitte Hoffmann, Hanaa Dahy

**Affiliations:** 1Research Group in Sustainable Design Engineering, Technical Faculty of IT & Design, Aalborg University, 2450 Copenhagen, Denmark; ursanoisabella@gmail.com (I.U.); lamiita0205@gmail.com (L.T.); 2Department of Planning, Technical Faculty of IT & Design, Aalborg University, 2450 Copenhagen, Denmark; bhof@plan.aau.dk (B.H.); hanaadahy@plan.aau.dk (H.D.); 3BioMat Department, Bio-Based Materials and Materials Cycles in Architecture, Institute of Building Structures and Structural Design (ITKE), University of Stuttgart, Keplerstr. 11, 70174 Stuttgart, Germany; 4Department of Architecture (FEDA), Faculty of Engineering, Ain Shams University, Cairo 11517, Egypt

**Keywords:** biomimicry, sustainability, sustainable design, sustainable innovation, sustainability promise, biologically inspired design, nature, mimesis, built environment

## Abstract

Biomimicry is an interdisciplinary approach to study and transfer principles or mechanisms from nature to solve design challenges, frequently differentiated from other design disciplines by its particular focus on and promise of sustainability. However, in the biomimicry and biologically inspired design literature, there are varying interpretations of how and whether biomimetic designs lead to sustainable outcomes and how sustainability, nature, and mimesis are conceptualised and engaged in practice. This paper takes a particular focus on the built environment and presents a theoretical overview of biomimicry literature spanning across specific fields, namely architecture, philosophy, sustainability and design. We develop upon conceptual considerations in an effort to contribute to the growing calls in the literature for more reflective discussions about the nuanced relationship between biomimicry and sustainability. We further develop a ‘Biomimicry for Sustainability’ framework that synthesises recent reflective deliberations, as a possible direction for further theorisation of biomimicry, aiming to elaborate on the role of biomimicry as a sustainable design methodology and its potential to cultivate more sustainable human–nature relations. The framework is used as a tool for retrospective analysis, based on literature of completed designs, and as a catalyst for biomimetic design thinking. The objective of this paper is to serve as a point of departure for more active and deeper discussions regarding future biomimetic practice in the context of sustainability and transformational change, particularly within the built environment.

## 1. Introduction

In the context of biologically inspired design (BID), biomimicry is an interdisciplinary approach, bringing together biologists, designers, engineers, among others, to study and transfer principles or mechanisms from nature to solve design challenges [1,2]. However, in the biomimicry and BID literature, there are varying interpretations of how and whether biomimetic designs lead to sustainable outcomes [2,3,4,5], how ‘nature’ is conceptualised within biomimetic practice, and to what extent ‘mimesis’ is a systematic transfer of nature or a discursive process of inspiration from nature [6,7,8]. Thus, we present a review of the biomimicry literature with a focus on the built environment, developing upon conceptual considerations, to contribute to the growing calls in the literature for more reflective discussions about the nuanced relationship between biomimicry and sustainability. Given the largely synonymous relation between the terms BID and biomimicry [9,10,11], we discuss the concept of biomimicry in the context of BID, which will be made clear in our use of design cases further below. Technical deliberations on the differences between biomimicry and BID are beyond the scope of the present paper as we aim to contribute to reflective discussions on biomimicry and its elusive role as a sustainable design methodology.

Biomimicry is framed as a response to the growing calls for alternatives to the ecologically destructive technologies, systems, and approaches of the current industrial age defining current unsustainable human–nature relations [8], which takes “nature as a model to meet the challenges of sustainable development” [12]. McDonough and Braungart’s Cradle to Cradle design approach reflects a similar orientation, maintaining that the laws of nature are the bedrock of good design [13]. In the literature, biomimicry is frequently characterised by its promise to achieve sustainable designs [2,3,14,15,16,17] (infrastructure design see [18], architectural and urban design see [16,19,20,21]), to reconnect humans with nature [1,4,7], to regenerate ecosystems [22], and to fundamentally transform the way we think by moving beyond anthropocentric narratives of human domination over non-human life [5,23,24].

The fundamental principles driving these biomimetic promises centre around the emulation of nature’s time-tested patterns. The argument builds on the fact that life on Earth has been developing for 3.8 billion years, while humans have been around for a mere 200,000 years—a blink of an evolutionary eye. Thus, nature is often conceptualised as “a catalogue of products” [7] (p. 6), “a living encyclopaedia of ingenuity” [1] (p. 122), “a large database of strategies and mechanisms” [20] (p. 693), or “a gigantic pool of ideas” [12] that has 3.8 billion years’ worth of insights and clever adaptations to offer—an untapped wisdom that humans should consult, emulate, and learn from to ‘achieve’ sustainability [7,19]. In other words, the natural world has characteristics that, if systematically studied and transferred, can help us move toward sustainability. Here sustainability is often treated as an achievable steady state for which biomimicry becomes a ‘tool’ or ‘vehicle’. In practice, designers translate this steady state in terms of harm reduction and damage limitation, thereby assuming an inherently harmful characterisation of human activity that must be minimised via optimal designs. However, this characteristic promise of sustainable designs is frequently contested in the literature, since biomimetic designs do not always lead to more sustainable outcomes [7,14,18,22,25], with the very notion of sustainability itself—a contested concept [26,27].

“*As a field, biomimicry is diverse and, at times, less than coherent. Its practitioners can scarcely agree on the term’s definition, on what level of fidelity to nonhuman life is required for a project to count as ‘biomimesis’ or to what ends its methods are best applied.*”[23] (p. 64).

“*Further critiques of biomimetics have noted that the application to sustainability challenges requires deeper levels of theorisation to have meaningful impact. The emerging literature on biomimetics, however, tends to be focused on the technical translation of biological function without explicit consideration for the state-of-the-art thinking regarding sustainability considerations*.”[5] (p. 2).

In response, we further propose a framework, which synthesises recent reflective deliberations in the literature, as a possible direction for further theorisation of biomimicry, aiming to elaborate on the potential of biomimicry to “help launch designers into their new role as sustainability interventionists.” [14] (p. 66) and to ultimately cultivate more sustainable human–nature relations.

In the following sections, we will provide an overview of these debates in the literature, preceded by an elaboration of our research methods, and bring to light the various classifications of biomimicry, and the fundamental assumptions underlying the conceptualisations of sustainability, nature, and mimesis in these discussions. Following this, we introduce the ‘Biomimicry for Sustainability’ framework that operationalises these considerations in order to elaborate on the potential of biomimicry and offer a possible direction for further theorisation of its application to sustainability challenges. Finally, we conclude with an open discussion regarding the further development of the framework, along with exploratory questions to guide future conceptual and experimental research agendas.

## 2. Materials and Methods

This study is based on secondary research and investigates the conceptual considerations of biomimicry from a large collection of research papers found in the following databases: EBSCOhost, Web of Science Core Collection and SpringerLink. The publications span across the fields of architecture, philosophy, sustainability and design. Keywords included sustainability, sustainable development, biomimicry, biologically inspired design, built environment sustainable design, and human–nature relations. The papers were chosen based on the following search criteria: recent publications (after 2015) as well as papers that explicitly discuss the nuanced relationship between biomimicry and sustainability (i.e., sustainability promise), methodological and conceptual considerations of the application of biomimicry as a design response to sustainability challenges, biomimicry as a novel way of thinking, and biomimicry as a paradigmatic shift in redesigning human–nature relations. The following review is thus based on over 30 papers that provided explicit definitions of biomimicry as well as reflective and critical debates on the promise of biomimicry to cultivate more sustainable human-nature relations. 

For the text mining and review process of the collected papers, we adopted the affinity diagram methodology, which involved the retrieval and recording of main points, followed by the identification of themes and patterns. The purpose in making use of these sources was to better grasp the complex and plural issues, promises and implications inherent to biomimicry as a concept. Relevant quotes defining biomimicry and its implications on design, sustainability, the conceptualisation of nature and mimesis, future promises and risks were extracted and recorded on a digital board. These were subsequently divided into groups, which were further combined into main themes, as shown in Figure 1 below, in order to identify possible patterns regarding biomimetic practice and sustainability. The end result brought clarity and consensus between different lines of thought and disciplines employing biomimicry in their work, which gave an informed and detailed foundation for the framework developed in the present research.

## 3. Results

In this section, we provide an overview of the various classifications of biomimicry identified in the literature, bringing to light the fundamental assumptions underlying the conceptualisations of sustainability, nature, and mimesis in these discussions. We subsequently introduce the ‘Biomimicry for Sustainability’ framework that operationalises these considerations in order to elaborate on the nuanced relationship between biomimicry and sustainability. 

### 3.1. Biomimicry for X’ Classification

In the literature, various classifications of biomimetic designs have been developed. Some are particularly technical, differentiating between ‘classes’ of bio-derived developments, such as Speck et al.’s decision tree classification tool [16]. This level of detail is beyond the scope of the present paper as it does not offer a relevant point of departure for our discussion regarding biomimicry as a sustainable design methodology. The classifications that will be discussed here are more broadly based on motivations for applying biomimetic approaches to the design process, i.e., a classification of biomimicry’s (design) promises [7,19,25]. This particular focus on the intentionality behind a design approach is in response to the notion of ‘naturalistic fallacy’ that is identified in the literature as a risk of biomimicry [7,8]. The naturalistic fallacy is committed when a design is argued to be ‘good’ or ‘sustainable’ because it is based on the principles of nature, i.e., it’s good because it’s natural.

Hence, in order to deepen the levels of conceptual considerations of biomimicry as a sustainability design methodology and avoid the naturalistic fallacy, we take the motivations driving biomimetic innovations as a point of departure. Common to many of the classifications based on biomimicry’s design promise is the distinction between what is called ‘biomimicry for innovation’ and ‘biomimicry for sustainability’ [7]. The former can be described by the study and application of nature’s evolutionarily optimised strategies through the interdisciplinary biomimetic process that is motivated by its particularly high potential for innovation, offering its practitioners a novel way of looking at design challenges without a driving concern for ecological performance [7,12,25]. ‘Biomimicry for sustainability’, on the other hand, is characterised by a more explicit concern to create designs that are well-adapted and integrated into life on Earth by accounting for product life cycles and Earth system limitations [14,25].

Within these articulations of biomimetic promises, sustainability is often treated as an achievable steady state for which biomimicry becomes a ‘tool’ or ‘vehicle’. In practice, designers translate this steady state in terms of harm reduction and damage limitation approaches with an explicit focus on discrete, easily quantifiable performance outcomes concerning improvements in efficiency during the use phase. For example, in the context of the built environment, biomimetic designs have led to reductions in embodied energy of construction materials, improvements in energy and structural efficiency as well as in material use and maintenance [1,20]. Such translations of the concept of sustainability are deeply rooted in ‘traditional’ articulations of sustainable development (e.g., net zero, harm reduction, damage limitation, eco-efficiency—see McDonough and Braungart [13] for an in-depth discussion of the notion of ‘eco-efficiency’ and “why being ‘less bad’ is no good”) that assume an inherently harmful characterisation of human activity that must be minimised.

More recently, however, such traditional framings of sustainability employed in biomimicry discussions have been broadened to encompass new articulations of how to be in the world that are based on an understanding of humans as a part of nature, rather than apart, echoing ongoing discussions of bio-inclusive ethics [4], conceptions of value [28,29,30], and net-positive outcomes [31,32]. Pedersen Zari, for example, proposes an additional category, ‘biomimicry for human psychological well-being’, which is rooted in the motivation to explore “whether design based on an understanding of the living world could contribute to increasing human psychological wellbeing” [25] (p. 18). Since her focus is based in architecture and urban design disciplines, her discussions of this third category centre around biophilic design, which combines research and frameworks of human psychological connection with the perceivable natural world [33,34] with spatial design and urban planning concepts [25].

MacKinnon, Oomen & Pedersen Zari [7] further develop on this, proposing the classifications, ‘biomimicry for transformation’ and ‘biomimicry for society’. In particular, they argue for the potential biomimicry has in presenting new narratives of sustainable human participation in nature and the realignment of human systems within biological systems. Although MacKinnon, Oomen & Pedersen Zari [7] do not differentiate between the two categories nor offer concrete translations of either in practice (i.e., what it means for the designer to pursue this ‘class’ of biomimicry), they do highlight a need for more reflective, than solely active, biomimetic practice in order to further develop the concept of biomimicry in an effort to realise its ambitious potential “to inspire new mindsets, values and narratives concerning the relationship between people and nature” [5] (p. 6). In the following section, we give a brief overview of the responses in the literature for these deeper levels of theorisation of the biomimicry concept that open up a space for more reflective discussions.

### 3.2. Strong vs. Weak Biomimicry’ Classification

In the literature, there are a growing number of responses that pursue this reflective approach. To illustrate this approach and the interpretive flexibility of the biomimicry concept, we will give an overview of the analytical classification of biomimicry that Blok and Gremmen [8] have formulated as ‘strong’ and ‘weak’ biomimicry, based on which they analyse biomimicry in terms of its conceptualisation of nature, mimesis, technology and ethics. 

As briefly touched upon above, within biomimetic practice, nature is often treated as a catalogue or database of products, ideas, and ingenuity for humans to categorise, disassemble and adapt to human systems. This conceptualisation of nature is characteristic of what Blok and Gremmen have categorised as “strong biomimicry” [8]. According to the authors [8], strong biomimicry assumes human epistemic sufficiency to ‘know’ nature and thus reproduce it in biomimetic designs. Consequently, the strong conception of mimesis hinges on the imitation of this perfect nature, by which elements of the natural world are “dissected, pulled apart and reconstituted as an assemblage of capacities” [23] (p. 73) to be studied, translated, and applied to human systems. At this categorical extreme, nature remains conceptualised as the first nature of Enlightenment thought—as a universal world ‘out there’ for human civilisation to work on and through [23]—an entity necessarily separate from the human realm. 

At the other extreme, the weak conception of biomimicry hinges on a less perfect nature upon which humans may build via a flexible understanding of mimesis that beckons “a sense of co-becoming, co-individuation of form and matter” [6] (p. 806). In other words, through weak mimesis, designers become drawn into a dynamic dialogue with material nature and “acknowledge that human beings are merely participants in rather than masters over a complex ontological entanglement from which emerges a shared design for (human and nonhuman) lived reality” [6] (p. 807). Similarly, MacKinnon, Oomen and Pedersen Zari characterise weak biomimicry as a process by which designers build upon natural inspiration with human analogical thinking, thus leaving “more room for flexibility in integrating nature into concrete biomimetic designs” [7] (p. 8). This categorical distinction reflects the nuanced articulations and conceptualisations of nature within biomimetic practice.

We argue that the more flexible conceptualisation of mimicry can allow a shift in the designer’s focus from performance outcomes to process outcomes, entering a more discursive and synergistic relationship with nature. In this way, the design of experimental and communicative processes can become an important element in developing what Freya Mathews calls ”a new culture of engagement with nature” by which we “allow nature to ‘redesign’ not only our commodities but also our own desires” [4] (p. 382). The ‘weak’ conceptualisation of biomimicry can thus encourage a more procedural understanding of sustainability that moves beyond the harm reduction discourse and shines a more positive light to human activity, calling for explorations of net-positive outcomes for both humans and non-human nature [27] and “conscious processes of learning and participation through action, reflection and dialogue” [35] (p. 678).

Although the ‘strong vs. weak’ discourse on biomimicry paints a relatively black and white classification of biomimicry, the debates are nuanced, indicating yet another potential for further analysis and expansion of the concept and its application. This nuanced view of biomimicry, however, is often overlooked and simplified in sustainable design literature. For example, Ceschin & Gaziulusoy’s Design for Sustainability framework [36] limits the scope of biomimetic design interventions primarily to the material, component, and product levels centred around user–product interactions. We see a clear underestimation of the potential of biomimicry to affect change on greater levels of socio-technical-ecological systems and human-nature interactions. Thus, we take this oversimplification as a point of departure for the development of a framework that aims to open up biomimicry as a design for sustainability methodology and actively bring its conceptual nuances to the forefront of biomimicry research. 

## 4. Analysis

In an effort to respond to this depreciation of biomimicry and highlight its broader role as a sustainable design methodology, we have attempted to frame our discussions above in terms of the ‘biomimicry for X’ classification and the nuanced conceptualisation of mimicry. Hence, the following discussion aims to present a possible direction for further development in constructing a more holistic and reflective theorisation of biomimicry and its application to sustainability challenges. It is important to highlight that the mapping of biomimetic designs within our framework is qualitative, thus it is subject to a certain degree of interpretation. 

### 4.1. First Dimension of the Biomimicry for Sustainability Framework

We position the ‘biomimicry for X’ classifications on our first dimension (x-axis), as depicted in Figure 2, which represents the scope of the biomimetic promise. This can also be understood in Pedersen Zari’s terms of “the projected end aspirations of different kinds of biomimicry [so as] to avoid the assumption that just because an object, material, system or building mimics nature in some way, it is inherently more sustainable” [25] (p. 17). Along the first dimension, we have placed design examples that are frequently mentioned in the literature [1,6,7,16,20] in order to illustrate each category. These designs were chosen to reflect the range of varying aims of biomimetic design, from techno-centric to biosynergistic objectives, as well as the diverse application of the biomimetic process.

In Figure 2, as we move further away from the origin to the right, the projected scope of the biomimetic intervention becomes increasingly concerned with responding holistically to sustainability challenges. We begin with ‘biomimicry for innovation’ on the far left, which encompasses biomimetic research and designs that “are about novel approaches to technical problems, increased performance capabilities, or the ability to increase economic profit margins’’ [25] (p. 17; for a more in-depth discussion of biomimicry as an avenue of innovation and economic production, by which nature remains entangled with logics of capital accumulation and resource privatisation, see [23]). This is a clear techno-centric aim whose projected end aspirations revolve around user-product interactions and commercial interests, like the example of Velcro that mimics the function of the way a seed from the Burdock plant attaches temporarily to an animal’s fur to travel long distances before germinating [37]. Also exemplified in the diagram is the Bullet Train, which mimics the form of the kingfisher’s beak that can move through air and water quickly and with minimum impact or noise. This example presents a novel approach to a technical problem (transportation) but is placed further to the right than Velcro given the design’s minimisation of air resistance and thus fuel-efficiency [38]. The last example placed within this category is Neri Oxman’s Silk Pavilion, which presents a novel approach to human problem (construction), with potential for lightweight structures [39,40].

Following is the ‘biomimicry for net-zero optimisation’ category, which presents a reformulation of the earlier discussion of ‘biomimicry for sustainability’, in order to reflect how sustainability is understood in terms of the net-zero approach that does not necessarily counteract unsustainable trajectories, but rather slows them down by optimising existing designs. The underlying assumption in this approach centres around a technical focus on reducing impact. The example placed in this section, ITKE’s Flectofin, is illustrative of a biomimetic design that has aimed for quantifiable performance outcomes such as reduced energy and material use [20,41]. It mimics the mechanism behind the movement of the bird of paradise flower when a bird lands on it, for adaptive exterior shading systems in buildings. The intention behind the design was the optimisation of energy consumption in mechanical cooling systems [20,41], hence its position within this second category.

Next, we move toward a more transformational conception of sustainability, based on which biomimicry is used to affect paradigmatic changes on a societal scale. Here, we coalesce ‘biomimicry for human psychological well-being’, ‘biomimicry for transformation’, and ‘biomimicry for society’, discussed above, under a more general classification that we call ‘biomimicry for societal transformation’, since all three present a particular objective to effectuate change on greater, societal levels through design. Here we offer the example of Yaniv Peer’s Mobius Project, which mimics ecosystem recycling of resources and aims to transform the role of urban spaces in terms of food production, waste management, community-building and education [1].

Within this category, the notion of biosphere limits and planetary boundaries functions as a fundamental principle driving responses to sustainability challenges. Research and design agendas centre around a knowledge-first approach, by which “science characterises problems in terms of their causes and mechanisms and forms a basis for subsequent action” [42] (p. 286). In this way, design work is driven by a concern to redesign our systemic means of fulfilling societal functions (e.g., transport, housing, food) that remain within the Earth’s carrying capacity. As the Mobius Project illustrates, the design does not challenge the amount of resources used or wasted in society, but rather offers a way to make the consequences of our societal actions less bad.

In line with Robinson & Cole [32] and Mathews [4,43], we argue that a shift beyond this ‘constraints and limits discourse’ is needed to engage in ‘a co-creative partnership with nature’ by designing processes of reflection, feedback and dialogue and thereby exploring possibilities of net-positive outcomes and synergistic human–nature relations. Borrowing Mathews’ notion of a ‘form of synergy proper to biomimicry’, we propose a fourth classification—‘biomimicry for biosynergy’. She defines ‘biosynergy’ as a novel direction in biomimicry thinking that shifts from “a mutualism of means [...] to a rapprochement of ends” [4] (p. 377). In other words, this new way of thinking shifts design intentions from fulfilling current human consumer ends in line with the interests of nature, to questioning our ends themselves—“what does the rest of nature want us to want?” [4] (p. 377).

Her notion of biosynergy thus takes us a step beyond net-positive sustainability and begs the question—from what perspective are net-positive outcomes positive? She argues that biomimicry must start from within nature by allowing “nature to design us as well as our instruments” [4] (p. 373). In this way, human activities can have a generative impact for nature, where nature is no longer a stable category of thought, distinct from the human realm. Here, the example of BioHaven’s Floating Islands presents a biomimetic innovation that transfers mechanisms from wetland ecosystems to improve water quality by capturing, absorbing or filtering organisms, chemical entities, etc., and to increase biodiversity by establishing micro-habitats [44]. This innovation illustrates an application of biomimicry that begins to approach Mathews’ [4] notion of biosynergy as its design has considered the ends of humans as well as nature. It is a clear example of a human innovation that has a positive generative impact on nature. However, the design of the islands does not necessarily intend to challenge current human–nature interactions that are causing the degradation of water systems. It does not motivate us to think about our ends themselves—how can we allow nature to redesign our desires?

### 4.2. Second Dimension of the Biomimicry for Sustainability Framework

In order to design from within ‘nature’s mindset’, we argue that the conception of mimesis is a helpful element to consider as it is pivotal in framing the designer’s relationship with nature. Thus, we introduce the second dimension (y-axis) to the framework, Figure 3, which spans from a ‘fixed’ to a ‘flexible’ conception of mimesis, bringing to analytical light the agency of nature within the design process. Although we take our point of departure in the ‘strong’ and ‘weak’ classification as identified in the literature, we reframe and re-articulate the ‘strong vs. weak’ distinction as a ‘fixed–flexible’ continuum. In this way, we avoid the normative connotation that the terms ‘strong’ and ‘weak’ carry, as reflected in the notions of strong and weak sustainability [45], thereby giving a more reflective space for such deliberations. Here, we take into consideration the roles of the human designer and non-human nature within the design process, where the fixed conception reflects an active role for the human designer and a passive role for nature, and the flexible conception implies a more active role for both. 

By introducing this second dimension to the framework, each mapped example is reassessed in terms of the conception of mimesis inherent to the particular design, and the kind of interaction between the designer and nature and is thus shifted vertically to be repositioned within the new range. This reassessment, however, requires an extensive review of the design process for each example that takes on a heightened analytical sensitivity towards this discussion of mimesis. In order to illustrate our framework, we present the model in Figure 3, in which we apply this approach to only two examples, which are discussed below, leaving the rest open for further deliberation. 

In particular, we shift Neri Oxman’s Silk Pavilion upwards due to the way she refers to this project as a way to reflect on how a “collaboration between us [humans] and these wonderful creatures [silkworms] could assist in structural optimization on a larger scale” [40]. The collaborative approach Oxman applied to her work is embodied in a designerly interaction, through which nature, in this case the silkworm, is given agency. In other words, both Oxman and the silkworms played active roles in the design process—their performances embedded in a materially discursive interaction with each other. We thus argue that her work embodies a more flexible conceptualisation of mimesis such that her design becomes an evolving process of thinking analogously with material nature, while the scope of the biomimetic promise remains within the more techno-centric realms.

The example of Flectofin, on the other hand, is shifted down as the design process employed a fixed imitation of nature, by which “a valvular pollination mechanism was derived and abstracted from the kinematics found in the Bird-Of-Paradise flower” for the development of a hingeless flapping device [41] (p. 1). Here, human epistemic sufficiency to ‘know’ and thereby reproduce nature is assumed, by which the design aim becomes the scientific identification, categorisation, abstraction and deployment of ‘natural design’ for the benefit of human civilisation [5]. In this example, the flower is a passive participant in the design process, whose contribution to the final design is actively mediated by the human designers. Thus, Flectofin is mapped as an example of ‘biomimicry for net-zero optimisation’ that employs a fixed conception of mimesis. 

Ultimately, we do not argue whether a fixed or flexible conception of mimesis is more or less likely to lead to sustainable designs. Rather, we argue that it offers a novel category of analysis, by which deliberations regarding the relationship between biomimicry and sustainability can further unfold in more transformational directions. Based on our review, however, there is a lack of design deliberations, experiments and developments that move toward the top, right corner of the diagram, since employing a flexible conception of mimesis and ideating biosynergistic designs are not straightforward tasks. Nevertheless, these challenges offer pathways for biomimetic design work to move toward continuous processes of reflexivity and learning about how we can realign ourselves with nature. 

It is important to note that, in this paper, the framework is used as a tool for retrospective analysis, based on literature of completed designs. For this reason, the analysis brought forward by the second dimension hinged on the materials of communication and the respective discursive understandings of the design processes. This extended our attention to the language employed in the literature, which refers to ongoing research in the ways in which biomimicry is communicated by practitioners (see [7]). Nevertheless, we maintain that the framework can also be used to stimulate critical reflections on the roles played by humans and nature and the characterisation of their interactions during a design process.

By turning an analytical eye toward the role and agency of nature within a biomimetic design process, practitioners can critically reflect on their designs and employ holistic and participatory perspectives on human–nature relations, in an effort to realise biomimicry’s promise of sustainability. We thus present this framework as a possible direction for further theorisation of biomimicry and its application to sustainability challenges. By constructing this two-dimensional plane, we hope to contribute to a more reflective space for emerging literature on biomimetics that aims to expand its focus on the technical translation of biological functions to also include explicit considerations of the ontological implications of biomimetics.

## 5. Discussion and Conclusions

This paper contributes to the understanding of the connections between biomimicry and sustainability, with a particular focus on the built environment, in an effort to motivate a more reflective approach to biologically inspired innovation. In response to the contested promise of sustainability within biomimicry discussions as well as the contested concept of sustainability itself, we provide an overview of these debates in the literature, bringing to light the various classifications of biomimicry and the fundamental assumptions underlying the conceptualisations of sustainability, nature, and mimesis. In order to operationalise the nuanced relationship between biomimicry and sustainability, we introduce a ‘Biomimicry for Sustainability’ framework that integrates and synthesises these debates, ultimately serving as a point of departure for more active and deeper discussions regarding future biomimetic practice. 

Regarding further development of the framework, alternative possibilities for the choice of dimensional ranges as well as the introduction of perhaps a third dimension can be considered. In particular, further considerations of scale can be taken in order to explore, for instance, possible correlations regarding the mimicry level of nature (i.e., form, process, ecosystem) as there is much debate in the literature surrounding the relationship between the various levels of biomimicry and sustainability promises. Pedersen Zari [22,25] and MacKinnon et al. [7], for example, argue that ecosystem level biomimicry leads to more sustainable designs, distinguishing between mimicry of ecosystem functions and ecosystem processes. Additionally, the scale or scope of the biomimetic innovation placed within the framework (i.e., component, product, service, system) can be investigated as an additional parameter for analysis regarding biomimetic designs as responses to sustainability challenges. Finally, the scope of human participation within the design process can be explored in order to stimulate conversations regarding transformational shifts within and across disciplines as well as society at large.

Along with the need of biomimicry’s philosophical elaboration and development, as identified in the literature and considered above, there are also calls for more experimentation, since little discussion is offered as to how these deeper elaborations of the concept can be applied in practice [3,4,6,23,43]. Active experimentation with the biomimetic process would further these conceptual developments by opening up a space for a form of methodological double-loop learning that positions biomimicry not only as the objective but as the object of study as well. Such experimentation and elaboration can be further complemented with literature, outside of the scope of biomimetic practice, from authors like Donna Haraway [46] and Anna Tsing [47], that bring to the forefront discussions of multispecies interactions, naturecultures, and interspecies relationships. In this way, biomimetic design experimentation can be situated as a catalyst practice for thinking through nature–culture dichotomies and driving sustainability transitions.

Before we can claim definitively whether biomimicry is an important key to delivering transformative change, we must consider the following questions, which we pose as input for future research agendas:

1. How can biomimicry be qualified in the development of sustainable design?

The Biomimicry for Sustainability framework (Figure 3) introduced in this paper offers a possible direction for future research in addressing this question—calling for the development of analytical tools to cultivate more reflective biomimetic practices.

2. How can a design process give agency to nature?

In other words, how can a flexible conception of mimesis be practically employed in a biomimetic process? In addition to reflective practices, future research should address experimental approaches that can expose a novel re-envisioning of the metaphysics of nature, which we argue, in line with Mathews [4,43], is key to sustainability. Such experimental practices, however, must also be considered in holistic terms, so that biomimicry can become a vehicle of change on larger, societal scales; thus, we pose the following question:

3. How can localised biomimetic design experimentation contribute to ongoing processes of common world-making and broader sustainability transitions?

Ultimately, responses of design disciplines, like biomimicry, to sustainability challenges must address the deepening entanglement of human and biosphere processes and the growing call to redesign human–nature relations. To do so, we maintain that conceptual deliberations and design experimentation are needed to actively reflect on the design aims and process as well as the roles of the designer(s) and nature. Thus, we have introduced the Biomimicry for Sustainability framework and posed the questions above to open up space for more active and deeper discussions regarding future biomimetic practice in the context of sustainability and transformational change.

## Figures and Tables

**Figure 1 biomimetics-07-00037-f001:**
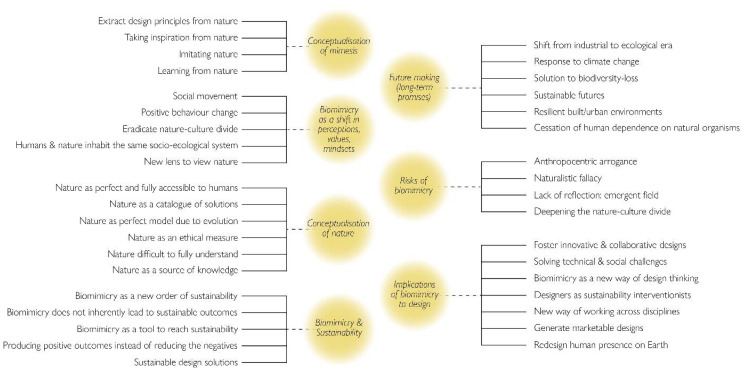
Resulting groups and themes from the affinity diagram process. The themes are represented by the yellow circles.

**Figure 2 biomimetics-07-00037-f002:**
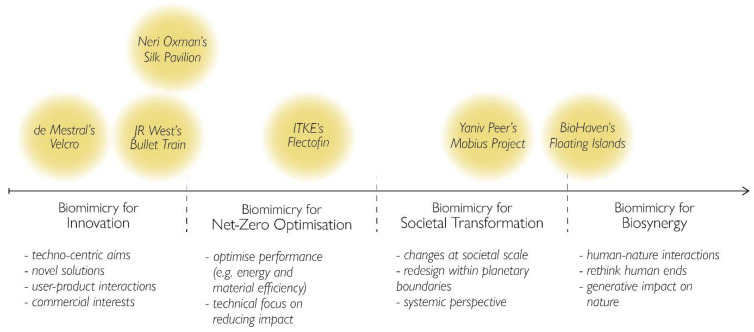
First dimension (x-axis) of the Biomimicry for Sustainability framework, with general criteria for each category. The varying placement of the examples in the vertical direction is arbitrary, for visual purposes only.

**Figure 3 biomimetics-07-00037-f003:**
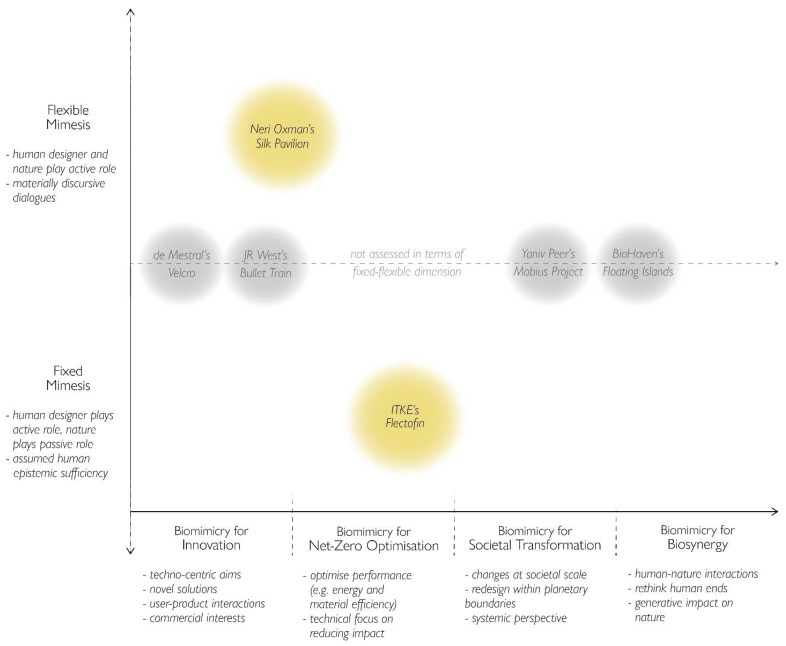
Biomimicry for Sustainability Framework with two examples (in yellow) assessed based on both dimensions. General criteria for each category included. The black and white examples have not been assessed in relation to the vertical range, as indicated by the dotted line.

## Data Availability

The data presented in this study are available in [48].

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
