# Peer review of "Biomimicry as a Sustainable Design Methodology—Introducing the ‘Biomimicry for Sustainability’ Framework"

_biomimetics, 2022, doi:10.3390/biomimetics7020037_

Round 1
Reviewer 1 Report
This is a well written and interesting paper. The introduction is well cited and provides a good context for the paper. The methods section is good. It would be helpful to have a graphic explaining the methods process, in particular the ‘large collection of papers’ that were started with and what the criteria was for the narrowing down to the 30 papers. It should be clarified what were the main groups and themes and how were these decided. The results section is good and well explained. In the analysis, the Figure 1 is a bit unclear as to why these particular projects were selected. This could be clarified. It is clear that the authors wanted to use examples from the built environment but these projects are very different from one another. How were they chosen? The discussion section is quite brief, and ending with the questions for future research seems abrupt. Perhaps there should be a final paragraph clarifying what these future research agenda questions could hope to achieve, what is the end goal. Overall a well written paper and a good contribution.
Reviewer 2 Report
The paper is suitable for the journal but would be also for Sustainability.
The abstract summarizes well the paper.
The paper is however not well categorized. From the materials and methods it is rather a review paper.
The Analysis section is original and good. There are strong points of originality in this.
The conclusions are completely absent and the reviewer suggestion is to rename the Discussion section to Discussion and conclusions.
Also the reviewer recommends to pay attention to formatting. There are sylabus divisions where there should not be and also colouring of some sections probably from copy-paste from a translation software.
Author Response
Please see the attachment.

This manuscript is a resubmission of an earlier submission. The following is a list of the peer review reports and author responses from that submission.
Round 1
Reviewer 1 Report
This paper is well written, compellingly argued and timely, particularly given the theme of 'how can we allow nature to redesign our desires?'.
Structurally I'm not sure that section 3.3 belongs in the Results section, as it is not the result of the literature search described in your Materials and Methods section, but rather your analysis of these findings.
Line 312: In line with Robinson & Cole [26] and Mathews [3,34], we argue that a shift beyond 312 this constrain[t]s and limits - remove t.
Line 323: beckoning the question should be 'begs the question'?
I don't understand why some of the examples in Figure 2 have not been reassessed and repositioned accordingly? Could you not just have a dotted line to show the original baseline?
The author of reference 1 should be :
-
Taylor Buck, N.
Reviewer 2 Report
This manuscript is more of a philosophical literature review other than original technical research. Although I do think a manuscript/discussion like this that reflects on the definition and framework of the term "biomimicry" is needed, and I applaud the efforts the authors made trying to tackle this topic and traverse in this murky water. Unfortunately, I did not find this manuscript to provide enough synergy and original insights to justify its publication. This manuscript also contains some outright errors that should not be present in the manuscript submitted for review, which are negative indicators for the quality of this manuscript. -- e.g., The "Error! Reference source not found." message in L139-140, L260, L275, L355, and L430.